# Reputation-Based Trust Approaches in Named Data Networking

**Ioanna Angeliki Kapetanidou** *,†, **Christos-Alexandros Sarros** † and **Vassilis Tsaoussidis** †

Department of Electrical and Computer Engineering, Democritus University of Thrace, 67100 Xanthi, Greece; csarros@ee.duth.gr (C.-A.S.); vtsaousi@ee.duth.gr (V.T.)
* Correspondence: ikapetan@ee.duth.gr
† Current address: Department of Electrical and Computer Engineering, Democritus University of Thrace, Kimmeria, 67100 Xanthi, Greece

**Abstract:** Information-Centric Networking (ICN) has arisen as an architectural solution that responds to the needs of today's overloaded Internet, departing from the traditional host-centric access paradigm. In this paper we focus on Named Data Networking (NDN), the most prominent ICN architecture. In the NDN framework, disseminated content is at the core of the design and providing trusted content is essential. In this paper, we provide an overview of reputation-based trust approaches, present their design trade-offs and argue that these approaches can consolidate NDN trust and security by working complementary to the existing credential-based schemes. Finally, we discuss future research directions and challenges.

**Keywords:** Named-Data Networking; Information-Centric Networking; reputation-based trust; security; trust

## 1. Introduction

Trust is a concept that is tricky to define. One of its definitions is "the extent to which one party is willing to participate in a given action with a given partner, considering the risks and incentives involved" [1]. A trust decision is based on the balance between trust and risk, and it has some sort of effect on the trustee.

Named Data Networking (NDN) [2], the most prominent ICN architecture, currently utilizes trust mechanisms that are based on cryptographic signatures and certificates. This approach is called credential-based [3] or policy-based [4] trust. It be considered as a "strong and crisp" approach, where decisions are founded on logical rules and verifiable properties encoded in digital credentials.

However, there exists another major approach to trust management, a "soft and social approach", based on reputation measures gathered and shared by a distributed community. This model is called "reputation-based trust". In this approach, when agents decide whether to trust another agent or not, they rely on evidence of past behavior supplied by trusted sources, as opposed to credentials.

In this paper, we provide an overview of reputation-based schemes designed for the NDN architecture and argue that reputation-based trust should also be considered as a viable, complementary solution for achieving trust in NDN, or other related architectures (NDN-based such as UMOBILE (A Universal, Mobile-Centric, and Opportunistic Communications Architecture) [5], or similar ones such as CCNx [6]).

Integrating reputation-based trust in NDN can also assist in securing environments that suffer from such issues, such as the Internet of Things (IoT). With the advent of the IoT and the new challenges that have arisen, many researchers have found in ICN an ideal match. The authors of Reference [7] argue that the IP model brings many constraints in such environments regarding

security and integration of local communications, introducing unnecessary complexity. On the other hand, retrieving named data in ICN naturally matches application semantics—simplifying their operation—while its main features can intrinsically support requirements such as energy efficiency, security, robustness, network scalability and mobility [8].

In the following sections, we first provide a background on reputation-based trust, as well as the NDN trust and security mechanisms currently in use, in Section 2. Then, we provide incentives for further investigating the potential of reputation-based trust in NDN and we overview the existing reputation-based mechanisms, in Section 3. Aiming to provide a generalized framework for designing reputation-based trust mechanisms in NDN, we detail the design options and their trade-offs in Section 4. Finally, we highlight the existing and potential use cases of reputation-based schemes in NDN and discuss related issues and challenges in Section 5.

## 2. Background

### 2.1. Reputation-Based Trust

In reputation systems, users assign ratings to service or resource providers. Those ratings represent a judgment of their direct interactions' quality. Eventually, trust is computed from the aggregation of ratings concerning local experiences, taking into account the feedback that is being provided by other network entities. A prominent example of a reputation-based system implementation is electronic commerce. E-markets, such as eBay and Amazon, have adopted reputation as a means for trust enforcement as well as an incentive to encourage transactions with unknown providers. On these platforms, reputation management is achieved through a centralized framework. This requires the existence of a trusted third party which collects the ratings and computes the final trust scores.

The deployment of trust management in commercial fields has triggered the extension to research in peer-to-peer (P2P) networking and multi-agent systems (MAS). The main feature of these networks that necessitates the existence of a mechanism for trust establishment, is the need of peers to identify if the intentions of other agents are reliable or malicious [9,10]. Contrary to online reputation models, the decentralized design of peer-to-peer networks and multi-agent systems requires a distributed approach. PeerTrust [11] proposes a distributed mechanism aiming to effectively evaluate the trustworthiness of peers and to detect various malicious behaviors in a P2P eCommerce community.

At the same time, trust has been investigated as an issue in Mobile Ad-hoc Networks (MANETs) and Wireless Sensor Networks (WSNs). In MANETs, multiple trust management schemes have been designed to address different goals, namely secure routing, authentication, intrusion detection, access control, key management and trust evidence [12]. More specifically, reputation-based approaches are typically used to secure routing, while credential-based approaches are preferred for the rest of the applicability domains. Characteristic of MANETs, is the fact that many trust-based protocols for secure routing derive trustworthiness from network-layer behavior and hence, networking parameters such as throughput or overhead are considered trust metrics.

WSNs have been another field of related research [13]. Similar to MANETs, trustworthiness in WSNs can be measured either by considering networking aspects (e.g., throughput, delay etc.) or by using typical reputation- and policy-based approaches. As reputation-based models require a reputation exchange protocol, they introduce significant communication overhead in resource-constrained WSNs, also impacting node energy efficiency. Their implementation in such environments is therefore challenging. One solution that has been explored in this context, is piggybacking reputation values on the routing packets. The trade-offs between the desired level of trust and resource availability in a wireless sensor network, demand that protocol design is tailored to the specific application requirements.

## 2.2. NDN Trust and Security

By making named data the central piece of its architecture, NDN secures the data itself, independent of the underlying communication channels. All NDN Data packets are immutable and each one carries a signature generated using its producer's cryptographic key at the time of data creation, binding its name to its content.

The NDN security framework is built on public-key cryptography. Applications and all other communication participants in an NDN network (entities), own one or more names. An entity proves its ownership of a name through an NDN certificate, which binds the name and a cryptographic public-private key pair possessed by the entity. Each entity can also issue certificates for the sub-namespaces it delegates to other entities. NDN security offers flexibility to application developers in deciding how to obtain certificates. Depending on the system design, a cloud-based application may obtain its certificate from a centralized certificate service, while a distributed application (e.g., P2P applications) may obtain the certificate from its users.

Utilizing public key cryptography also requires NDN to establish trust anchors. NDN necessitates that all cryptographic verifications must terminate at a pre-established trust anchor, while assuming that the authority of each networked system establishes its own trust anchor(s) and that all the entities under that authority can discover these trust anchors through local system settings. This trust model resembles that of the Simple Distributed Security Infrastructure (SDSI/SPKI) in trust anchor establishment.

By using the aforementioned mechanisms, NDN applications can validate received data packets independent from where they are fetched, while utilizing name semantics to reason about which cryptographic keys to use for which content, instead of relying on the "yes-or-no" model provided by third-party certificates [14].

In this context, it is clear that trust has been mainly addressed in NDN by using credential-based trust mechanisms. The main research focus has been on utilizing public key cryptography to validate communications and addressing related challenges, such as establishing trust anchors, providing effective solutions for trust management and developing usable key management solutions [15,16]. To achieve this, the NDN security framework utilises 3 main components: Digital Keys, Certificates and Trust Policies. By combining the above, NDN ensures Data authenticity, integrity and, optionally, confidentiality at a fine granularity.

Signature verification is performed by consumers for each Data packet, by retrieving the certificate of its producer. This certificate represents its issuer's endorsement of the binding between the name and the public key. It is a Data packet carrying a public key and can be fetched like any other Data packet. The issuer will put its signing key name with other auxiliary information into Data's signature info field.

The authenticity and integrity of received Data packets (some of them may be certificates) are determined by a combination of the following two factors: Validation of trust policies and Signature verification. When fetched, a certificate recursively points to its signer's certificate and finally arrives at a specified trust anchor. The Data packet is considered valid only if all the certificates in the above chain have valid signatures and satisfy the trust policies [14].

In NDN, applications define trust policies that specify which entities are trusted for producing which piece of data, as well as which key should be used for which data namespace (and for what purpose). Therefore, different trust models are used by different applications and trust context is defined by the application. Users and applications can express their trust policies in a form that can be directly executed by applications, by using so called "trust schemas" [16].

In contrast to the built-in security features, access control is not inherent in NDN architecture. This implies that content providers are responsible for designing and implementing related access control methods in order to protect their content from unauthorized consumers. To this end, utilizing credential-based schemes, and particularly encryption-based techniques [17], is a common approach.

For instance, the main concept of the mechanism proposed in Reference [18] is to generate two data packets for each content object—a first one including the encrypted content and a second including

a related access policy. An access policy is defined as a set of attributes which are used to infer if a user is allowed to access specific content. In this model, in order for a user to obtain a desired content object, the user must not only maintain a secret key to decrypt the content but the user's attributes should also match the attributes included in the associated access policy (specified in the second data packet), as well. A centralized proxy server (e.g., a selected router) is delegated with the responsibility to check if users' attributes match those specified by the corresponding access policy and thus, inspect access to protected content.

## 3. Reputation-Based Approaches in NDN

### 3.1. Motivation

Most of the existing NDN security problems stem from the fact that NDN routers must support structures such as a Pending Interest Table (PIT) and a Content Store (CS). The Content Store and Pending Interest Table enable in-network caching and aggregation of closely spaced interests for popular content, which are key features of the NDN architecture. Each router maintains a CS, which is a temporary cache of received popular content. When an Interest packet arrives at an NDN router, the router first checks its Content Store for matching data; if the data exists, it is retrieved and the Interest packet is immediately satisfied. If the matching Data packet is not available in the CS, the router turns to the Pending Interest Table. The PIT stores all the Interests that have been forwarded but not yet satisfied, as well as all related interfaces. When checking the PIT, if a matching entry exists, the router simply records the incoming interface in the PIT entry. Otherwise, it creates a new PIT entry, adds the incoming interface and forwards the Interest to the network (based on information provided by the Forwarding Information Table (FIB)). Later, when the matching Data packet is found and arrives at the router, it gets cached in the CS and forwarded to all recorded downstream interfaces in that PIT entry.

The CS and the PIT structures enhance system performance, as they significantly reduce the overall latency and improve bandwidth utilization for popular content. However, despite their obvious benefits, they also trigger new security issues as they make the network susceptible to new forms of DoS attacks.

A first type of such attacks, which targets the router's PIT, is called Interest Flooding Attack (IFA). When an Interest Flooding Attack is launched, an adversary floods a victim's PIT with Interests for non-existent content. Those malevolent Interests fill up the router's PIT and remain there until their expiration time. As a result, Interests issued by legitimate users are dropped, degrading network QoS. Interest Flooding Attacks are considered as a major security problem in NDN [19,20]. Most popular countermeasures involve some form of pushback mechanism (e.g., Reference [20]), but researchers have gone as far as proposing to eliminate the PIT structure in order to mitigate such attacks [21].

Another type of attacks aim at compromising the router's Content Store. The most common cache-related attacks are cache pollution and cache poisoning, while there are also less popular attacks such cache snooping. On a cache snooping attack, the adversary's goal is to gain complete knowledge of a cache's content. To achieve this, the attacker sends multiple Interests for various name prefixes and analyzes the time needed for the Interests to be satisfied. This attack usually targets edge routers to which a limited number of consumers is attached. Whenever the attacker succeeds, the content that the router's adjacent consumers are interested in, can be inferred. Therefore, the consumer's privacy is being violated by the attacker.

In a cache pollution attack, an attacker sends multiple requests for unpopular content and misleads the routers into filling up their caches with unpopular content. This violation of content locality in the caches results in cache misses for benign users and increased network traffic, as legitimate Interests have to be forwarded to the network as the cannot be answered by the router caches.

When it comes to cache poisoning attacks, the adversary anticipates genuine interests for a valid content name and injects fake content under the same name into router caches. The term "fake content" refers to content that is either signed with an invalid signature or signed with a valid signature which

is verified by an invalid public key [22]. While poisonous content travels back to the issuer of the Interest, it is cached by the routers along the path. To make things worse, subsequent Interests for the same content are satisfied by the infected CS and the fake content is spread to the network.

In the case of a cache poisoning attack, the adversary exploits the lack of any in-network signature verification. In the NDN/CCN architecture, consumers are obliged to verify the signatures of received content, while routers are not mandated to do so. There are three main reasons for this: (1) the computational cost of in-network cryptographic operations such as signature verification, can be considered prohibitive at line speed (at least for high-speed networks), (2) an additional overhead is being introduced in order to fetch the associated certificate chain, and (3) there is no unified trust model in NDN [23].

Regarding signature verification, it is considered as a heavy cryptographic operation, especially when compared to signature generation [24,25]. NDN currently offers two signature algorithms by default, RSA and ECDSA. Based on previous results, the authors of Reference [26] argue that stretching existing state of the art signatures (and their computationally expensive cryptographic primitives, such as modular exponentiations or elliptic curve point multiplications) so as to perform wire-speed verification over line-rates in the order of 10 or 100 Gbps is hard, to say the least. Results in Reference [27] showed that an optimized software implementation of RSA1024 signature verification running on Intel Core 2 Duo 2.53 GHz CPU allows a router to verify about 150 Mbps of traffic, assuming 1500 Bytes per content packet, or even worse with smaller-sized packets. As a result, the authors conclude that NDN routers with multiple Gigabit-speed interfaces would need an unrealistic amount of computing power to verify signatures of Data packets at wire speed.

A more recent study [28] focused on the per-packet verification cost and showed that the default NDN signature verification algorithm requires 0.0839 ms for verification of a 1000 bytes content packet on an Intel Core 2 1.83 GHz processor, using MD5 as the message-digest algorithm, RSA1024 as the asymmetric cryptographic algorithm, and DES-XEX3/CTR as the symmetric cryptographic algorithm. This amounts to 11,918 packet signature decryptions per second, far from meeting Gbps forwarding requirements. While another study [24] calculated that the average time for signature verification ranges from 1.274 ms (RSA algorithm) to 6.869 ms (Ring signature), on an Intel $^®$ Core$^{TM}$ i7-7700 CPU @3.6 GHz, 16 GB RAM desktop computer (packet size is not specified).

What is more, the results in Reference [29] show that the performance impact by signature verification may be high even on end-user client nodes, as throughput decreased by 5 to 20 times on web browsers running NDN.JS when signature verification was enabled. The authors contribute the severity of the decrease on the fact that JavaScript is not optimized for computation-intensive tasks, which made signature verification the bottleneck in content processing.

Equipping routers with Bloom Filters so that they perform mainly lookup operations and probabilistic signature verification (as proposed in access control models [30,31]) could bypass signature verification for each Data packet by all routers (e.g., by storing the namespaces of identified attackers), but additional issues remain.

Besides the excessive cost of signature verification, routers would need to fetch multiple public key certificates in order to trust the public key that verifies a content signature, which increases the overhead significantly. Furthermore, each content-producing NDN application can independently specify the trust model it utilizes. Therefore, there are diverse trust policies defined by different applications at the same time. This means that each router should know the specific trust schema used by each NDN application. Since there is a wide variety of applications in a network, this requirement entails considerable trust-related complexity for routers.

As involving routers in the specifics of trust management is cumbersome, routers have little defense means to mitigate cache poisoning attacks. In this light, reputation-based trust schemes have been leveraged as an alternate method to cope with cache poisoning. In addition, due to their light-weight design, reputation-based models have also been investigated as a means for mitigating other attacks, such as IFAs.

## 3.2. Reputation-Based Systems

When a cache poisoning attack is detected, the compromised router should discard the forged content from its Content Store. NDN inherently supports no other way to flush content from router caches than cache replacement policies, for example LRU. However, those policies rely solely on eventual natural cache aging and are not useful for distinguishing valid and poisonous content.

The authors of Reference [22] propose a reputation-based technique which is based on explicit exclusion filters to decrease the detected poisonous content in the caches. In particular, Interest packets include an optional field which is called "Exclude". This field contains information about name components that must not occur in the name of the returned content. The main concept of the proposed mechanism is that consumers, as they typically verify signatures, can detect invalid content and issue a new Interest to exclude certain content by referring to its hash (i.e., by exploiting the Exclude field). Routers rank the cached content based on the exclusion information provided by consumers, assigning a higher rank to valid content than to fake ones. The rank of a content object is based on the exclusion rate, the time freshness of the exclusions and the total number of interfaces which excluded this content and indicates how trusted the content is.

Although this algorithm achieves efficient mitigation of cache poisoning and delivery of valid content to consumers, it is still susceptible to these attacks. On the one hand, malicious end-users can target a valid content object and collaborate to exclude it from caches. On the other hand, a group of malicious consumers can explicitly request fake content and not exclude it. Therefore, poisonous content will be prioritized by routers and is likely to serve subsequent interests, propagating into the network. Another drawback of this method is that every content object, either valid or fake, will be cached, increasing memory cost.

The aforementioned vulnerabilities are considered important by Reference [32], whereby a different approach exploits excluding information, as well. The proposed reputation-based method assigns not only a trust value to each content object, but also a credibility value to each content provider. A user's credibility is computed based on the trust value of the content that was provided by the user. When a consumer issues an Interest excluding fake content, the content's trust value will be decreased, impacting the credibility of the sender, as well. Even though credibility values will be significantly decreased in case a poisonous content is detected, they will recover slowly by sending valid content. This technique aims at preventing on-off attacks, whereby adversaries send alternatively valid and invalid data to maintain a certain credibility value.

The calculated trust values and the credibility values are used by routers as criteria to perform selective caching and Interest acceptance. More particularly, a router caches received content with a probability which is equal to its trust value. In addition, routers accept the incoming Interests with a probability which is equal to the credibility value of the consumer who sent the Interest. Authors argue that associating the probability that a consumer's Interests are satisfied with their credibility value, incentivize users to provide valid content. Nonetheless, it is important to note here that the exclusion functionality exploited by the above-described mechanisms is no longer available in NDN since version 0.3, rendering the previous methods infeasible.

Cache poisoning attack mitigation is examined from a different angle in Router-Oriented Mitigation (ROM) [33]. ROM is based on the idea that content-oriented mitigation methods might be inadequate if the router itself tampers content passing through. In ROM, every router assigns a reputation value to each of its adjacent routers. Authors argue that the further a router is from an attacker, the more poisoned copies are received for a specific content piece because of multi-path propagation. Based on this assumption, they use the number of received poisoned copies of a certain content as a means to decide how much to decrease the reputation of routers that forward fake content. Reputation values are updated based on negative verification results generated by consumers and forwarded back to the routers following the reverse transmission path. This is similar to Reference [32], although the reputation value of a malicious router is significantly decreased when the router is punished, it takes time for it to recover.

The computed reputation value of a router represents how trusted this router can be and determines how likely it is for this router to be included in the transmission path. In other words, routers choose their well-reputed neighbors as next hops to forward Interests and thus, malicious routers are temporarily excluded from the transmission path.

ROM's efficiency is evaluated by comparing it with the Interest-Key Binding (IKB) rule [23], which states that an interest must reflect the public key of the producer. More specifically, IKB exploits an Interest field called PublisherPublicKeyDigest (PPKD). PPKD contains the SHA-256 (Secure Hash Algorithm-256)digest of the publisher public key. According to this method, the public key of every received content is being hashed by routers and compared to the PPKD of their related PIT entry. The content object is forwarded and cached by routers and its signature is verified by consumers if and only if the hash and the PPKD match. Otherwise, it is discarded.

However, since this approach is credential-based, it comes with its flaws. A consumer that issues an Interest has to fetch the PPKD of the desired content's provider in advance. In addition, the PPKD has to be verified at every hop. As these features increase the delay and reduce the network throughput, ROM outweighs IKB. In particular, evaluation results provided in [33] indicate that ROM improves network latency by 85.5% and throughput by 84%.

Regarding cache snooping, Reference [34] proposes a reputation-based method to detect adversaries. As explained in Reference [34], a snooper not only requests but even excludes content so that he can have a full picture of the cached content. Based on this assumption, the authors consider a user malicious when high interest and exclusion rates are measured in a short time period, as well as a high cache hit rate from the local cache at the same time. These features are used in the calculation of a user's reputation and subsequently, in the evaluation of another metric, the user's trust value. The trust value is compared against a threshold in order to detect snoopers.

Besides cache-related attacks, reputation-based schemes are used as a means to mitigate Interest Flooding attacks. ICRP (An Interest flow control method based on user reputation and content name prefixes ) [35] assigns a reputation value to each consumer which represents the transmission degree of Interests requiring existing content objects. Aiming at preventing on-off attacks, while calculating reputation values, ICRP weighs both the past and the current behavior of users. Computed reputation values are compared against a predefined threshold, and if a user's reputation is below this threshold, he/she is considered malicious. Upon the detection of a malicious consumer, its Interests are accepted according to its reputation value. The lower the reputation value, the higher the Interest drop rate. Moreover, ICRP observes and counts the Interests sent by detected attackers and creates a blacklist which includes non-existent name prefixes. Therefore, ICRP identifies the malicious users based on reputation values of consumers, while it detects and records non-existent content names and thus, limits the flow of malevolent Interests in the network.

Based on the evaluation results provided by the authors, reputation-based approaches have been shown as a pragmatic solution to efficiently mitigate cache- and PIT-centric attacks, and consolidate the existing credential-based NDN security schemes. Table 1 focuses on certain related works' efficiency evaluation results.

**Table 1.** Related work efficiency evaluation.

| Reputation-Based Scheme | Evaluation Metrics | Results |
| --- | --- | --- |
| Ghali et al. [22] | Percentage of benign consumers receiving valid content | 100% (after 5–60 s) |
| Rezaeifar et al. [32] | False Positive Error Ratio (FPER)<br>False Negative Error Ratio (FNER) | 21%–25%<br>10%–14% |
| Wu et al. [33] | False Positive Error Ratio (FPER)<br>False Negative Error Ratio (FNER) | <= 4%<br>0% |
| Umeda et al. [35] | Average data acquisition rate | > 70% |

## 4. Design Options

Depending on the implementation domain of a reputation-based trust management scheme, there are various design options which need to be examined. In this section, we explore the design space by laying out all options and assessing the trade-offs of each choice.

While analyzing the use cases in which the reputation-based scheme will be deployed, it is vital to examine three basic dimensions, namely formulation, calculation and dissemination, so that a comprehensive framework is tailored [36]. The formulation dimension describes the mathematical model and input for the assessment of reputation values. Formulation includes two main aspects: the reputation measure and the mathematical model used to aggregate ratings. Reputation can be measured using binary, discrete, continuous, string or vector type of values. The aggregation model can be a simple summation or average of ratings, discrete, fuzzy logic, flow-based or probabilistic [37].

**Content rating versus Network entity rating (Figure 1):** A basic dilemma in an NDN trust management scheme is whether the feedback (i.e., ratings or reputation value) should be tied to the disseminated content itself or to the entity which forwards the content (e.g., initial producer or intermediate router). The policy-based approach in force, relates content objects to certificates. Each time a pending Interest is satisfied, the requesting consumer has to verify the signature of the providing party, and thus accept or reject the incoming Data Packet. Managing trust that way, ensures resistance to attacks at consumers and can be utilized to exclude malicious content from intermediate router's caches [35].

Nonetheless, using a credential-based approach, consumers and intermediate routers cannot make any proactive assumptions regarding the trustworthiness of providers, or their future content quality. Introducing a reputation-based approach can have two benefits for NDN operation. On the one hand, when reputation values are assigned to providers, they can function as a criterion to (proactively) avoid transactions with untrustworthy providers, or perform selective forwarding and/or caching. On the other hand, assessing content quality by itself can be useful for consumers or for detecting unexpected or malevolent behaviors (e.g., malfunctioning sensors sending corrupted/false data).

Even though rating content has been investigated as an option for either ranking already cached content [22] or to determine the probability of content to be cached [32], the majority of previous works assume that ratings are assigned to network entities (e.g., end-users [32,34,35] or routers [33]).

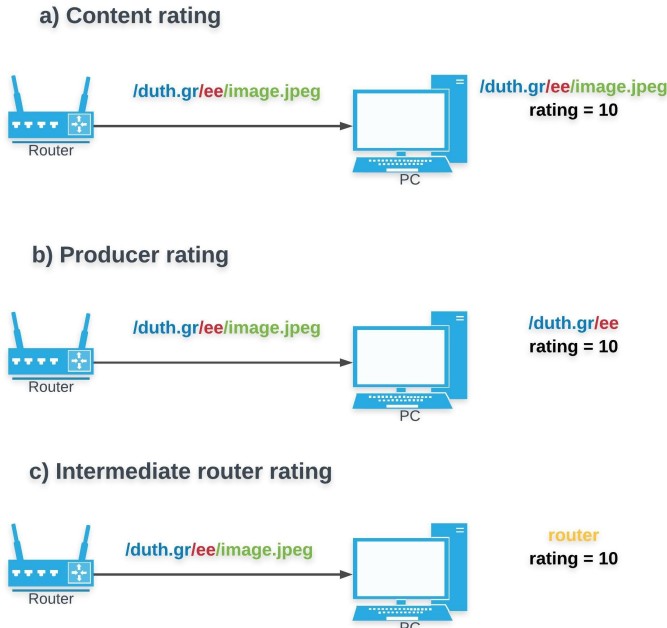

**Figure 1.** (**a**) Content Rating vs (**b**) & (**c**) Network Entity Rating.

Regarding reputation calculation and dissemination, the calculation dimension addresses the practical design and implementation of the reputation algorithm, whereas the dissemination dimension focuses on the mechanisms used for distribution and storage of ratings and reputation values among entities within the system. Based on classifications [38] and measures [37] for reputation-based schemes, we highlight below the ones that are most relevant for NDN.

**Centralized versus Decentralized (Figure 2):** A primary design decision regards the process through which reputation is being calculated and/or distributed in the network. Based on this, trust models are divided into centralized or decentralized ones. In the centralized case, a trusted central authority undertakes the responsibility to collect the ratings, compute the reputation values, store and announce them to network entities. Although this method limits computational complexity, since there is no need for every node to evaluate reputation, there is a trade-off with the introduced overhead for ratings' and reputation values' exchange. Another important drawback of this approach is that the single trusted entity represents a single point of failure. Once this is compromised, the entire network operation can fail.

On the other hand, there are decentralized approaches, whereby ratings are calculated by each node independently and then may be (optionally) distributed between the nodes by a dissemination protocol, in a P2P fashion. This approach eliminates the single point of failure problem, but requires that the individual nodes performing the calculations and providing the reputation values can be trusted, as well. To solve these problems, blockchain technology could be integrated in NDN as a solution, as it offers both the trustworthiness a central authority would (e.g., by recording ratings in the ledger, making them immutable and verifiable), combined with the distributed announcement of reputation values.

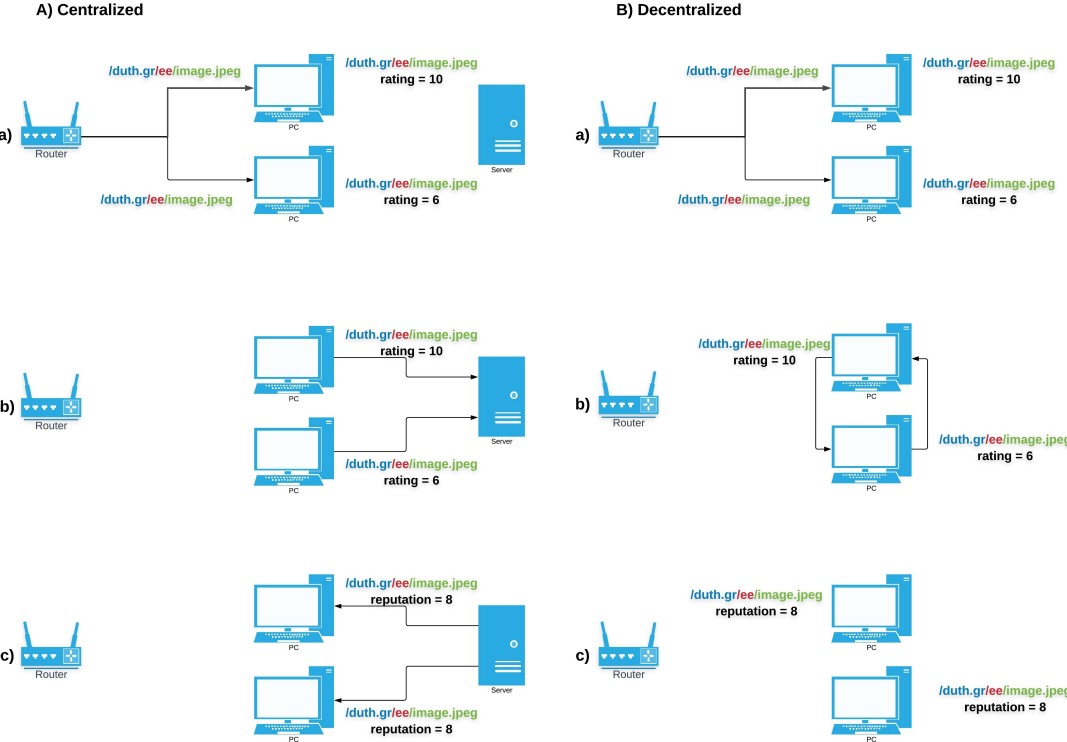

**Figure 2.** Centralized vs Decentralized: (**A**) Centralized: (**a**) Receiving content & assigning ratings, (**b**) Sending ratings to a CA, and (**c**) CA computing & announcing reputation value. (**B**) Decentralized: (**a**) Receiving content & assigning ratings, (**b**) Exchanging ratings, and (**c**) Computing reputation value individually.

**Transaction-based versus Opinion-based trust (Figure 3):** The former relies on objective information derived from transactions. Verification of data integrity and provenance authentication can be considered as trust metrics, as they are supplied by NDN's default trust management scheme. Alternatively, networking parameters such as delay could as well be a viable option. The latter, opinion-based trust, is based on the subjective opinion of an entity for another based on their past interactions. Although opinion-based mechanisms come with a lower overhead, they are more vulnerable to attacks. As a result, transaction-based schemes are preferred in security-related domains while opinion-based are most suitable when the main objective is data quality assessment.

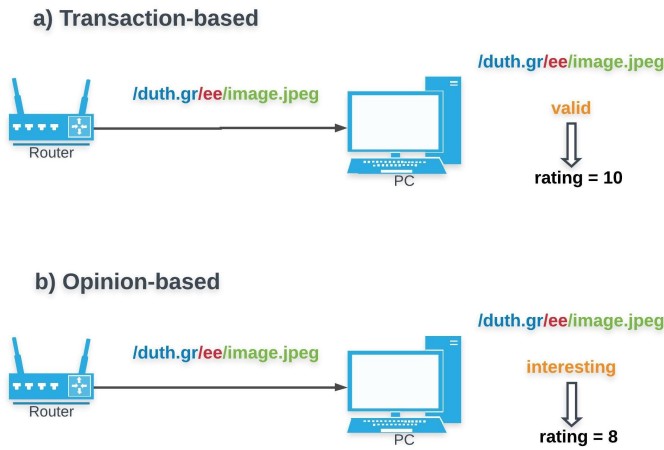

**Figure 3.** (**a**) Transaction-based vs (**b**) Opinion-based.

**Global reputation versus Local reputation (Figure 4):** Reputation can either be seen as a global property shared by all, or as a subjective property assessed locally by each entity. The localized approach is privacy-preserving and more suitable for decentralized environments. However, the global approach produces more accurate results as the calculation uses a more sufficient, representative amount of ratings. In NDN, global availability of reputation values can complement attack mitigation, (e.g., cache-poisoning). A local approach is more appropriate for assessing the quality of data.

It is worth highlighting here that all aforementioned works propose decentralized mechanisms which compute trust locally. In addition, ratings are assigned based on information derived from completed transactions. In particular, some rely on exclusion information to provide ratings [22,32], while others quantify the reputation values by evaluating the number of received poisoned copies for certain content [33], or by taking into account measurements of Interest [35] and exclusion [34] rates.

**Rank versus Threshold based schemes (Figure 5):** Rank-based approaches return a relative ranking of all entities' reputation. NDN can thus prioritize content with higher quality or providers with higher reputation. Such an approach has been used to prioritize valid over fake content [22] or to favor the most trustworthy routers to be selected as next hops during forwarding [33]. However, rank-based methods come with their drawbacks. For instance, high-ranked content is not always the best match for an Interest. In contrast, a threshold-based approach excludes entities which do not meet the desired criteria. Due to its strictness, it can be more efficient when leveraged to make forwarding and caching decisions [32,35] or to detect an attack [34,35].

**Incentive- versus Rule-based control (Figure 6):** A rule-based system forces entities to follow its rules (e.g., rating according to specified parameters). An incentive-based ones, motivates entities to follow desired behaviors (e.g., provide ratings). In contrast to [32], which associates a consumer's credibility with the probability that his/hers Interests are satisfied and exploits it as an incentive for users to provide valid content, the rest of the methods [22,33–35] follow specific rules.

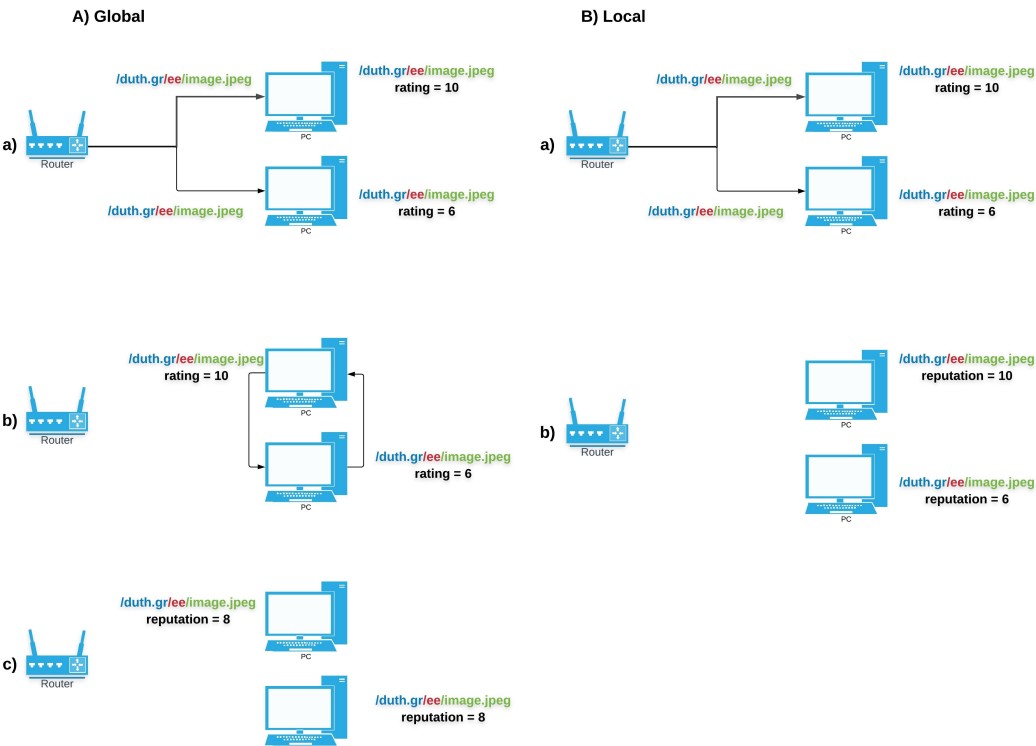

**Figure 4.** Global vs Local reputation: (**A**) Global: (**a**) Receiving content & assigning ratings, (**b**) Exchanging ratings, and (**c**) Computing reputation value individually. (**B**) Local: (**a**) Receiving content & assigning ratings, (**b**) Computing reputation value locally.

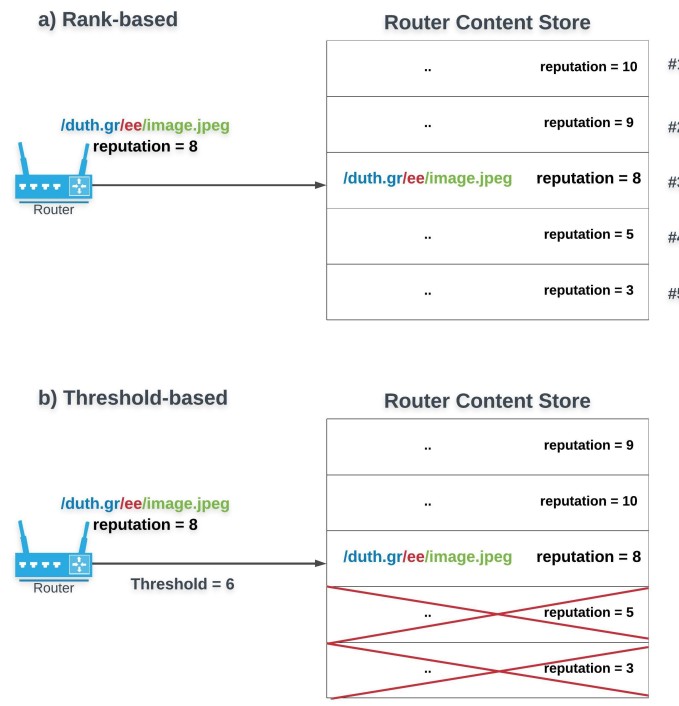

**Figure 5.** (**a**) Rank-based vs (**b**) Threshold-based.

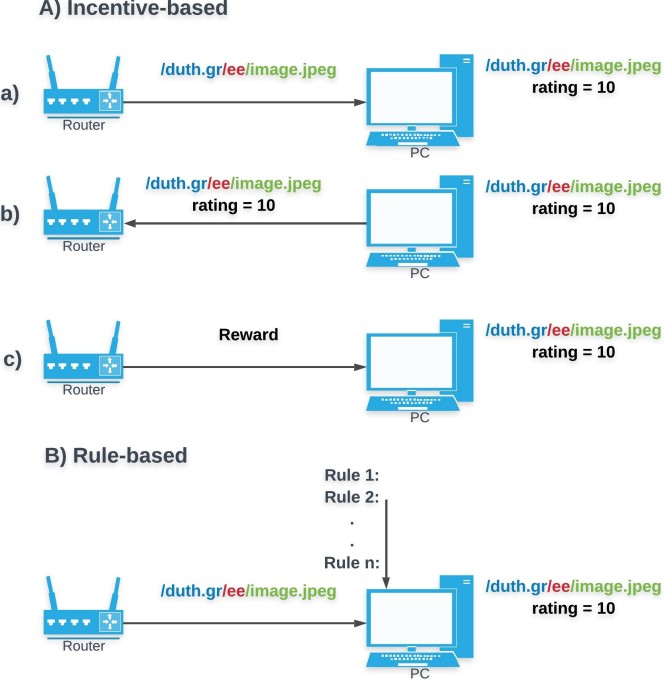

**Figure 6.** Incentive-based vs Rule-based: (**A**)Incentive-based: (**a**) Receiving content & assigning rating, (**b**) Propagating rating to the network, and (**c**) Getting reward, (**B**) Rule-based: Receiving content and assigning rating based on predefined rules.

Besides the aforementioned main design options, there are also secondary ones.

**Data Filtering versus Data Aging:** Data filtering favors a subset of the ratings by assigning a higher weight factor. This selected subset includes ratings based on measures, for example time freshness. Data aging assigns an aging factor to ratings. As information gets older, its weight in the calculation diminishes, and eventually it gets discarded. Assigning a higher weight factor to latest measurements can allow for faster detection of misbehaviors. The main drawback of these approaches is that they introduce complexity in reputation calculation.

**Collection:** It refers to the difference between information sources taken into account for the calculation of ratings and reputation values. Direct experiences and witness information are typical information sources used by reputation-based models [12]. These options have to be compared in the case of a localized scheme, since a global approach requires sharing the complete information range among entities.

In Table 2 we provide a mapping of the proposed NDN reputation-based mechanisms to the design decisions their authors have made (where the "*" symbol indicates that the related option has been chosen).

**Table 2.** Related work classification.

|  | Ghali et al. [22] | Rezaeifar et al. [32] | Wu et al. [33] | Ntuli et al. [34] | Umeda et al. [35] |
|---|---|---|---|---|---|
| Content rating | * | * | | | |
| Network entity rating | | * | * | * | * |
| Centralized | | | | | |
| Decentralized | * | * | * | * | * |
| Transaction-based | * | * | * | * | * |
| Opinion-based | | | | | |
| Global reputation | | | | | |
| Local reputation | * | * | * | * | * |
| Rank-based | * | | * | | |
| Threshold-based | | * | | * | * |
| Rule-based | * | | * | * | * |
| Incentive-based | | * | | | |

## 5. Discussion

### 5.1. Future Directions in NDN

Based on the existing literature, there two distinct *use cases* of reputation-based schemes which have been explored in the NDN architecture.

**Security Attacks:** Reputation-based methods have proven to be an efficient way to mitigate attacks in NDN, based on previous work. More specifically, reputation is considered useful in order to ensure protection against cache poisoning and Interest Flooding attacks. As analyzed above, given the fact that credentials and cryptography are prohibitively expensive techniques to deploy at routers, the latter are left susceptible to cache poisoning and IFAs. Reputation-based schemes appear to be a lightweight means to cope with the aforementioned threats efficiently and thus, this seems a promising research direction.

**Privacy Attacks:** Another research direction which exploits reputation in order to shield routers' caches and has been little explored, are cache snooping attacks. As cache snooping aims to the breach of consumers' privacy by inspecting the edge routers' cache content, reputation can be leveraged to detect snoopers and prevent them from violating legitimate users' privacy.

Nonetheless, it is important to highlight that reputation-based schemes are neither fully explored in the above use cases, nor restricted to them. Reputation can be leveraged to assist in trust management in other research directions, where it has not been investigated yet, as well. Such potential directions are:

**Bitrate Oscillation Attacks (BOAs):** In adaptive streaming, a client, based on measurements from the last delivered segment (e.g., assessment of the available bandwidth), decides which resolution of the next segment to request. Authors in [39,40] propose a novel attack model implemented in dynamic adaptive multimedia streaming. In this model, an attacker requests non consecutive segments of a content, causing them to be cached in intermediate routers' Content Stores. When a legitimate user subsequently requests the same content, the already requested segments are delivered to him from intermediate caches while the rest are sent by the initial producer. In his attempt to adapt to currently measured network conditions, a client is being misled into making false estimations. As a result, the victim is forced to experience bitrate oscillations and degraded QoE.

A reputation-based system can be designed in order to evaluate and detect unusual users' behaviors or to assess a priori if a content piece can be exploited by an attacker to perform a BOA. In the dynamic adaptive streaming (DAS) protocol, a user before obtaining the segments of the desired multimedia content, has to fetch a Media Presentation Description (MPD) file related to this content.

The MPD file includes information regarding the characteristics of the multimedia content's segments' (e.g., available bitrates, resolutions, codecs etc). Using a reputation-based mechanism, routers could rate the content based on the information included in the received MPD files. For example, they can assign reputation values to content according to the difference between the segments' available bitrates. In this case, reputation values will express how strong potential oscillations can be. Aiming at mitigating BOAs in NDN, reputation values can be exploited as condition for caching in order to accomplish oscillation avoidance and hence, weaken the attacker.

**Secure Routing:** Reputation can also be useful for routers, enabling them to make appropriate forwarding decisions. Routers can assign ratings to their adjacent routers and choose to which they should forward Interests based on their neighbors' trustworthiness. In a decentralized framework, reputation values should be synchronized. This requires developing a related exchange protocol which introduces undesirable communication overhead. In order to minimize this overhead, reputation values can be piggybacked in routing messages [12]. This direction has only been substantially explored in wireless networks but appears to be a promising feature for NDN routing protocols [41], as well.

**Content/Service quality evaluation:** Besides complementing NDN's credential-based methods to assuring that invalid content is discarded from the network, reputation can provide a second layer of trust in NDN. Similar to ratings in P2P networks, reputation values can allow consumers to reason about content quality, instead of simply certifying its authenticity and integrity. Activating this functionality can possibly be useful to address another challenge, as well, which is creating client profiles based on end-users' preferences and making those profiles available to content providers (e.g., Netflix) [42,43]. NDN enables bringing reputation directly on top of the network layer, possibly utilizing cross-layer information, a direction which has not yet been explored. An extension of this direction could be further explored for Service-Centric Networking approaches [44–46], the reputation being based on service quality, availability and other metrics.

*5.2. Open Issues & Challenges*

Based on the aforementioned merits of reputation-based schemes, we argue that all the above directions need to be further investigated. Different design option combinations, which result to different performance trade-offs, have to be thoroughly explored and NDN performance needs be quantified.

For example, reputation values can be tied to each content object. This is a fine-grained approach, but introduces high overhead as the number of different ratings that would have to be stored, computed and disseminated is proportionate to the number of content objects. Furthermore, no reasoning can be made about a producer's future content. Reputation values can also be mapped to producers, performing some form of rating aggregation (based on the ratings on each producer's content objects). This enables pro-active reasoning about the trustworthiness of a producer's future content, but may also harm legit new objects (fake positives). Furthermore, there are technical challenges, for example how to extract the producer from a Data packet. A solution is to aggregate all ratings to the previous level of the naming hierarchy, the other one being to rate the associated certificate's identity. This method exploits the NDN's certificate format, which requires certificates names to adhere to the format /<SubjectName>/KEY/[KeyId]/[IssuerId]/[Version]. What is important, is the fact that a certificate name starts with the identity to which a public key is bound, that is, the namespace under which it can sign Data packets. Therefore, trust ratings can be provided for each associated identity to assess the trustworthiness of its certificates and published content.

Another open issue is how to authenticate the ratings. As NDN is designed with consumer-oriented privacy in mind, the Interest packets are not signed by default. An option is to providing the ratings using signed Interests, but, in this case, consumer privacy would be violated and routers would be made vulnerable to flooding attacks by signed Interests (which would be much more effective than typical Interests, due to the signature verification overhead). The other option is to design a pull-based protocol, so that the ratings are provided by Data packets which are always signed. But this would

dictate that consumers cannot immediately provide a rating for each Data packet they receive, instead they have to wait for their ratings to be periodically requested and collected (the higher the frequency, the larger the overhead).

Furthermore, a reputation-based system relies on consumer feedback. Deployable solutions need to ensure that this feedback is being provided. A transaction-based system design can hard-code this in protocol design, for example when a signature cannot be verified, the protocol is going to provide negative feedback. Clearly, this is an option for mitigating attacks. On the other hand, an opinion-based system that focuses on data quality assessment, may require end-user participation as well as proof that a transaction has taken place. This would also which necessitates an mechanism for such a proof, as well as an incentives mechanism. Blockchain-based solutions can be used in this case, such as Proof-of-Prestige [47] which forms a reward system for unverifiable tasks.

## 6. Conclusions

Achieving trust in NDN has mainly relied on credential-based mechanisms. In this paper, we provided an overview of the existing work on reputation-based trust and explored their design options, trade-offs and prospects. We argue that the existing research shows potential and that this direction needs to be further explored by the ICN community.

**Author Contributions:** All authors contributed equally to the conceptualization and the writing of the paper.

**Funding:** This research was partly funded by the Greek State Scholarships Foundation (IKY) action "Supporting human research resources through doctoral research", Operational Programme "Developing Human Resources, Education and Lifelong Learning", 2014–2020 and the Seventh Framework Capacities "Research Potential" program of the European Union grant number 264226, FP7-REGPOT-2010-1

**Conflicts of Interest:** The authors declare no conflict of interest.

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
