# Peer review of "Reputation-Based Trust Approaches in Named Data Networking"

_futureinternet, doi:10.3390/fi11110241_

Round 1

Reviewer 1 Report

The authors study the problem of reputation-based trust approaches for named data networking. They discuss the state of the art in NDN and present the different reputation schemes presented in the NDN literature and other well-known reputation schemes. The authors discuss reputation based approaches presented in NDN for mitigating cache pollution attacks, cache poisoning attacks, and also comment on router based mitigation efforts. 

The reviewer has the following suggestions:

The authors should consider citing a very comprehensive survey in ICN security, privacy, and access control and contrast their paper in terms of the difference in their paper.  Tourani, Reza, et al. "Security, privacy, and access control in information-centric networking: A survey." IEEE communications surveys & tutorials 20.1 (2017): 566-600. The exclusion based approaches presented by the authors should come with a caveat that the exclusion functionality is no longer available in NDN. The authors state that the assessment of the packets by the routers is expensive. Probabilistic assessment is not that expensive. The authors should mention the following papers (particularly paper 3) in this paper:

T. Chen, K. Lei, and K. Xu, “An encryption and probability based access control model for named data networking,” in Proceedings of IEEE International Performance Computing and Communications Conference, 2014, pp. 1–8. 

R. S. D. Silva and S. Zorzo, “An access control mechanism to ensure privacy in named data networking using attribute-based encryption with immediate revocation of privileges,” in Proceedings of IEEE Consumer Communications and Networking Conference, 2015, pp. 128–133. 

Tourani, Reza, Ray Stubbs, and Satyajayant Misra. "TACTIC: Tag-Based Access ConTrol Framework for the Information-Centric Wireless Edge Networks." 2018 IEEE 38th International Conference on Distributed Computing Systems (ICDCS). IEEE, 2018.

The authors also talk about there not being a mechanism for eliciting client preference in NDN, the authors should cite the following papers in the area and discuss the relevant proposed add-ons in this paper.

Ó Coileáin, Diarmuid, and Donal O'Mahony. "SAVANT: Aggregated feedback and accountability framework for named data networking." Proceedings of the 1st ACM Conference on Information-Centric Networking. ACM, 2014.

Tourani, Reza, Satyajayant Misra, and Travis Mick. "Application-specific secure gathering of consumer preferences and feedback in ICNs." Proceedings of the 3rd ACM Conference on Information-Centric Networking. ACM, 2016.

The authors need to a better job of integrating the suggested solutions into NDN. At the moment, the proposed solutions appear superficial and would benefit from more thinking of the solutions.    

More edits in attached file.

Author Response

Response to Reviewer's 1 Comments

The authors study the problem of reputation-based trust approaches for named data networking. They discuss the state of the art in NDN and present the different reputation schemes presented in the NDN literature and other well-known reputation schemes. The authors discuss reputation based approaches presented in NDN for mitigating cache pollution attacks, cache poisoning attacks, and also comment on router based mitigation efforts.

The reviewer has the following suggestions:

Point 1: The authors should consider citing a very comprehensive survey in ICN security, privacy, and access control and contrast their paper in terms of the difference in their paper.  Tourani, Reza, et al. "Security, privacy, and access control in information-centric networking: A survey." IEEE communications surveys & tutorials 20.1 (2017): 566-600.

Response 1: Added a new paragraph in “NDN Trust & Security” subsection:
“In contrast to the built-in security features, access control is not inherent in NDN architecture. This implies that content providers are responsible for designing and implementing related access control methods in order to protect their content from unauthorized consumers. To this end, utilizing credential-based schemes, and particularly encryption-based techniques [17], is a common approach.”

Point 2: The exclusion based approaches presented by the authors should come with a caveat that the exclusion functionality is no longer available in NDN.

Response 2: Added a new paragraph:“Nonetheless, it is important to note here that the exclusion functionality exploited by the above-described mechanisms is no longer available in NDN since version 0.3, rendering them infeasible.”

Point 3: The authors state that the assessment of the packets by the routers is expensive. Probabilistic assessment is not that expensive. The authors should mention the following papers (particularly paper 3) in this paper:

T. Chen, K. Lei, and K. Xu, “An encryption and probability based access control model for named data networking,” in Proceedings of IEEE International Performance Computing and Communications Conference, 2014, pp. 1–8.

R. S. D. Silva and S. Zorzo, “An access control mechanism to ensure privacy in named data networking using attribute-based encryption with immediate revocation of privileges,” in Proceedings of IEEE Consumer Communications and Networking Conference, 2015, pp. 128–133.

Tourani, Reza, Ray Stubbs, and Satyajayant Misra. "TACTIC: Tag-Based Access ConTrol Framework for the Information-Centric Wireless Edge Networks." 2018 IEEE 38th International Conference on Distributed Computing Systems (ICDCS). IEEE, 2018.

Response 3: Added the following paragraphs in “NDN Trust & Security” subsection:

“In contrast to the built-in security, access control is not inherent in NDN architecture. This implies that content providers are responsible for designing and implementing related access control methods in order to protect their content from unauthorized consumers. To this end, utilizing credential-based schemes, and particularly encryption-based techniques [17], is a common approach.

For instance, the main concept of the mechanism proposed in [18] is to generate two data packets for each content object: a first one including the encrypted content and a second including a related access policy. An access policy is defined as a number of attributes which are used to infer if a user is allowed to access specific content. In this model, in order for a user to obtain a desired content object, the user must not only maintain a secret key to decrypt the content but the user's attributes should match the attributes included in the associated access policy (specified in the second data packet), as well. A centralized proxy server (e.g. a selected router) is delegated with the responsibility to check if users' attributes match those specified by the corresponding access policy and thus, inspect access to protected content.”

Added a new paragraph in “Motivation” subsection:
“Equipping routers with Bloom Filters so that they perform mainly lookup operations and probabilistic signature verification (as proposed in access control models [30,31]) could bypass signature verification for each Data packet by all routers (e.g. by storing the namespaces of identified attackers), but additional issues remain.”

Point 4: The authors also talk about there not being a mechanism for eliciting client preference in NDN, the authors should cite the following papers in the area and discuss the relevant proposed add-ons in this paper.

Ó Coileáin, Diarmuid, and Donal O'Mahony. "SAVANT: Aggregated feedback and accountability framework for named data networking." Proceedings of the 1st ACM Conference on Information-Centric Networking. ACM, 2014.

Tourani, Reza, Satyajayant Misra, and Travis Mick. "Application-specific secure gathering of consumer preferences and feedback in ICNs." Proceedings of the 3rd ACM Conference on Information-Centric Networking. ACM, 2016.

Response 4: Integrated this in the “Content/Service quality evaluation” future direction description.
Original text: “Besides complementing NDN's credential-based methods to assuring that invalid content is discarded from the network, reputation can provide a second layer of trust in NDN. Similar to ratings in P2P networks, reputation values can allow consumers to reason about content quality, instead of simply certifying its authenticity and integrity. NDN enables bringing this feature directly on top of the network layer, possibly utilizing cross-layer information, a direction which has not yet been explored. An extension of this direction could be further explored for Service-Centric Networking approaches [42-44], the reputation being based on service quality, availability and other metrics.”

New text: “Besides complementing NDN's credential-based methods to assuring that invalid content is discarded from the network, reputation can provide a second layer of trust in NDN. Similar to ratings in P2P networks, reputation values can allow consumers to reason about content quality, instead of simply certifying its authenticity and integrity. Activating this functionality can possibly be useful to  address another challenge, as well, which is creating client profiles based on end-users' preferences and making those profiles available to content providers (e.g. Netflix) [42,43].

NDN enables bringing this feature directly on top of the network layer, possibly utilizing cross-layer information, a direction which has not yet been explored. An extension of this direction could be further explored for Service-Centric Networking approaches [44-46], the reputation being based on service quality, availability and other metrics.”

Point 5: The authors need to a better job of integrating the suggested solutions into NDN. At the moment, the proposed solutions appear superficial and would benefit from more thinking of the solutions. 

Response 5: In order to provide an example of how reputation-based trust could be integrated in an NDN framework, we provide an initial design of a reputation-based mechanism for Bitrate Oscillation Attack mitigation:

“A reputation-based system can be designed in order to evaluate and detect unusual users' behaviors or to assess a priori if a content piece can be exploited by an attacker to perform a BOA. In the dynamic adaptive streaming (DAS) protocol, a user before obtaining the segments of the desired multimedia content, has to fetch a Media Presentation Description (MPD) file related to this content. The MPD file includes information regarding the characteristics of the multimedia content's segments' (e.g. available bitrates, resolutions, codecs etc). Using a reputation-based mechanism, routers could rate the content based on the information included in the related received MPD files. For example, they can compute the difference between the segments' available bitrates and assign reputation values to content according to it. In this case, reputation values will express how strong potential oscillations can be. Aiming at mitigating BOAs in NDN, reputation values can be exploited as condition for caching in order to accomplish oscillation avoidance and hence, weaken the attacker."

Reviewer 2 Report

The work is well written and makes an interesting analysis regarding attacks and reputation solutions in NDN. However, the work could be improved if the authors turned it into a survey. This way, the authors could add more references and make more comparisons between different methods. 

Ln 84-87. Mentioning that there is no need for a third party to manage the public key at this point in the text may lead the reader to a misunderstanding. I suggest a detailed paragraph to avoid misinterpretation.

Ln 95-99. This paragraph could be rewritten to detail in depth the relationships involving public keys and how they would be securely delivered to the consumer. 

Ln 151-152. "the computational cost of any in-network cryptographic operation is prohibitive at line speed"   

It would be interesting to substantiate this statement by showing results that prove it. 

Ln 172. Rewrite this sentence: “Authors of [21] propose a reputation-based technique to decrease the of detected poisonous content in the caches.” 

Ln 180. Rewrite this sentence :"and it indicates how trusted the content is," 

Ln 222-224. Please provide more detail. It is not clear if the signature is verified only by consumers or if there is a feedback to inform routers to remove from the cache. 

Ln 227-228. Please provide more detail about the measures taken to calculate the throughput reduction. 

Ln 310-317. The authors could reference more works and make a comparison. 

Reduce the size of Figure 5 and Figure 6. 

In addition to Table 1, the authors could make more comparative tables in terms of attack containment efficiency and the computational cost involved. 

Author Response

Response to Reviewer's 2 Comments

Point 1:The work is well written and makes an interesting analysis regarding attacks and reputation solutions in NDN. However, the work could be improved if the authors turned it into a survey. This way, the authors could add more references and make more comparisons between different methods.

Response 1: Reputation-based trust as an attack mitigation means has been explored only limitedly in NDN. This implies that existing literature is rather restricted and, to the best of our knowledge, we have covered it to a considerable extent. Having taken into account this limitation, our purpose is mainly to provide an overview of the existing reputation-based approaches in NDN and argue that research towards this direction should be made, rather than to present a comprehensive survey.

Point 2: Ln 84-87. Mentioning that there is no need for a third party to manage the public key at this point in the text may lead the reader to a misunderstanding. I suggest a detailed paragraph to avoid misinterpretation.

Response 2: Rephrased the text, by adding two paragraphs:

“By making named data the central piece of its architecture, NDN secures the data itself, independent of the underlying communication channels. All NDN Data packets are immutable and each one carries a signature generated using its producer's cryptographic key at the time of data creation, binding its name to its content.

The NDN security framework is built on public-key cryptography. Applications and all other communication participants in an NDN network (entities), own one or more names. An entity proves its ownership of a name through an NDN certificate, which binds the name and a cryptographic public-private key pair possessed by the entity. Each entity can also issue certificates for the sub-namespaces it delegates to other entities. NDN security offers flexibility to application developers in deciding how to obtain certificates. Depending on the system design, a cloud-based application may obtain its certificate from a centralized certificate service, while a distributed application (e.g., P2P applications) may obtain the certificate from its users.

Utilizing public key cryptography also requires NDN to establish trust anchors. NDN necessitates that all cryptographic verifications must terminate at a pre-established trust anchor, while assuming that the authority of each networked system establishes its own trust anchor(s) and that all the entities under that authority can discover these trust anchors through local system settings. This trust model resembles that of the Simple Distributed Security Infrastructure (SDSI/SPKI) in trust anchor establishment.

By using the aforementioned mechanisms, NDN applications can validate received data packets independent from where they are fetched, while utilizing name semantics to reason about which cryptographic keys to use for which content, instead of relying on the "yes-or-no" model provided by third-party certificates [14].”

Point 3: Ln 95-99. This paragraph could be rewritten to detail in depth the relationships involving public keys and how they would be securely delivered to the consumer.

Response 3: Initial text: “Signature verification is performed by a consumer for each Data packet, by retrieving the certificate of its producer (identified by the key name in a dedicated packet section). This certificate recursively points to its signer’s certificate and finally arrives at a specified trust anchor. The Data packet is considered valid only if all the certificates in the above chain have valid signatures and satisfy the trust policies.”

Rewritten as:
“Signature verification is performed by a consumer for each Data packet, by retrieving the certificate of its producer. This certificate represents its issuer’s endorsement of the binding between the name and the public key. It is a Data packet carrying a public key and can be fetched like any other Data packet. The issuer will put its signing key name with other auxiliary information into Data’s signature info field.

The authenticity and integrity of received Data packets (some of them may be certificates) are determined by a combination of the following two factors: Validation of trust policies and Signature verification. When fetched, a certificate recursively points to its signer’s certificate and finally arrives at a specified trust anchor. The Data packet is considered valid only if all the certificates in the above chain have valid signatures and satisfy the trust policies.”

Point 4: Ln 151-152. "the computational cost of any in-network cryptographic operation is prohibitive at line speed" 
It would be interesting to substantiate this statement by showing results that prove it.

Response 4:
- Rephrased to “the computational cost of in-network cryptographic operations such as signature verification, can be considered prohibitive at line speed (at least for high-speed networks)”, to be more precise.

- Added the next paragraphs that elaborate on the previous sentence, including references [23-28].

“Regarding signature verification, it is considered as a heavy cryptographic operation, especially when compared to signature generation [24,25]. NDN currently offers two signature algorithms by default, RSA and ECDSA. Based on previous results, the authors of  [26] argue that stretching existing state of the art signatures (and their computationally expensive cryptographic primitives, such as modular exponentiations or elliptic curve point multiplications) so as to perform wire-speed verification over line-rates in the order of 10 or 100 Gbps is hard, to say the least. Results in [27] showed that an optimized software implementation of RSA1024 signature verification running on Intel Core 2 Duo 2.53 GHz CPU allows a router to verify about 150 Mbps of traffic, assuming 1,500 Bytes per content packet, or even worse with smaller-sized packets. As a result, the authors conclude that NDN routers with multiple Gigabit-speed interfaces would need an unrealistic amount of computing power to verify signatures of Data packets at wire speed.

A more recent study [28] focused on the per-packet verification cost and showed that the default NDN signature verification algorithm requires 0.0839 ms for verification of a 1000 bytes content packet on an Intel Core 2 1.83 GHz processor, using MD5 as the message-digest algorithm, RSA1024 as the asymmetric cryptographic algorithm, and DES-XEX3/CTR as the symmetric cryptographic algorithm. This amounts to 11,918 packet signature decryptions per second, far from meeting Gbps forwarding requirements. While another study [24] calculated that the average time for signature verification ranges from 1.274 ms (RSA algorithm) to 6.869 ms (Ring signature), on an Intel® Core™ i7-7700 CPU @3.6 GHz, 16BG RAM desktop computer (packet size is not specified).

What is more, the results in [29] show that the performance impact by signature verification may be high even on end-user client nodes, as throughput decreased by 5 to 20 times on web browsers running NDN.JS when signature verification was enabled. The authors contribute the severity of the decrease on the fact that JavaScript is not optimized for computation-intensive tasks, which made signature verification the bottleneck in content processing.”

Point 5: Ln 172. Rewrite this sentence: “Authors of [21] propose a reputation-based technique to decrease the of detected poisonous content in the caches.”

Response 5: Rewritten (“the of detected” -> “the detected”)

Point 6: Ln 180. Rewrite this sentence: "and it indicates how trusted the content is,"

Response 6: Replaced the comma (,) with a full stop (.)

Point 7: Ln 222-224. Please provide more detail. It is not clear if the signature is verified only by consumers or if there is a feedback to inform routers to remove from the cache.

Response 7: Initial text:
“According to this method, the public key of every received content is being hashed and compared to the PPKD of the related PIT entry. The signature is verified by consumers and the content is forwarded and cached if and only if the hash and the PPKD match. Otherwise, the content is discarded.”

Rewritten as:
“According to this method, the public key of every received content is being hashed by routers and compared to the PPKD of their related PIT entry. The content object is forwarded and cached by routers and its signature is verified by consumers if and only if the hash and the PPKD match. Otherwise, it is discarded.”

Point 8: Ln 227-228. Please provide more detail about the measures taken to calculate the throughput reduction.

Response 8: Added two figures. The first one shows a comparison between ROM’s and PPKD’s latency (proving that ROM results in reduced delay), while the second one presents a comparison in terms of throughput (proving throughput increment when using ROM).

Point 9: Ln 310-317. The authors could reference more works and make a comparison.

Response 9: Due to the restricted existing literature on reputation-based approaches in NDN architecture, including more references was not a trivial task. Besides that, based on your suggestion, we integrated the proposed reputation-based mechanisms (described in the related work section) in the design options analysis, as well.  This integration concerns not only the two last design options (ln 310-317), but the whole section and aims at making the comparison between the related works’ design choices more profound. Furthermore, we have added a new Table (Table 1), in which we compare between the efficiency of different reputation-based trust approaches in NDN.

Point 10: Reduce the size of Figure 5 and Figure 6.

Response 10: Resized Figures 5 and 6.

Point 11: In addition to Table 1, the authors could make more comparative tables in terms of attack containment efficiency and the computational cost involved.

Response 11: Added a new table (new “Table 1” - old “Table 1” is now “Table 2”) in “Reputation-based Systems” subsection which provides a comparison in terms of efficiency evaluation results.

Round 2

Reviewer 1 Report

The authors have adequately addressed the reviewers concerns.